# Task dynamics define the contextual emergence of human corralling behaviors

**Patrick Nalepka**[1,2]*, **Paula L. Silva**[3], **Rachel W. Kallen**[1,2], **Kevin Shockley**[3],
**Anthony Chemero**[3], **Elliot Saltzman**[4,5], **Michael J. Richardson**[1,2]

**1** School of Psychological Sciences, Macquarie University, Sydney, NSW, Australia, **2** Centre for Elite Performance, Expertise and Training, Macquarie University, Sydney, NSW, Australia, **3** Department of Psychology, Center for Cognition, Action & Perception, University of Cincinnati, Cincinnati, OH, United States of America, **4** Department of Physical Therapy & Athletic Training, College of Health & Rehabilitation Sciences, Sargent College, Boston University, Boston, MA, United States of America, **5** Haskins Laboratories, New Haven, CT, United States of America

* patrick.nalepka@mq.edu.au

**Data Availability Statement:** The data collected in the human experiment can be publicly accessed at Open Science Framework https://osf.io/w4bae/. The corralling task environment and the software

## Abstract

Social animals have the remarkable ability to organize into collectives to achieve goals unobtainable to individual members. Equally striking is the observation that despite differences in perceptual-motor capabilities, different animals often exhibit qualitatively similar collective states of organization and coordination. Such qualitative similarities can be seen in corralling behaviors involving the encirclement of prey that are observed, for example, during collaborative hunting amongst several apex predator species living in disparate environments. Similar encirclement behaviors are also displayed by human participants in a collaborative problem-solving task involving the herding and containment of evasive artificial agents. Inspired by the functional similarities in this behavior across humans and non-human systems, this paper investigated whether the containment strategies displayed by humans emerge as a function of the task's underlying dynamics, which shape patterns of goal-directed corralling more generally. This hypothesis was tested by comparing the strategies naïve human dyads adopt during the containment of a set of evasive artificial agents across two disparate task contexts. Despite the different movement types (manual manipulation or locomotion) required in the different task contexts, the behaviors that humans display can be predicted as emergent properties of the same underlying task-dynamic model.

## Introduction

Social animals have the extraordinary capacity to structure their activity in coordination with other members of a larger group. The resultant behaviors that emerge at the collective level display key features of self-organizing systems [1], whereby interactions among individuals give rise to functionally organized, coordinated behavioral patterns. These patterns show a remarkable degree of qualitative similarity across species [2–6]. For example, similar herding and containment (i.e., corralling) behavior has been observed during group hunting by wolves [7] and certain cetaceans [8–10] despite the different biological and environmental constraints acting

used for the model simulations, as well as software to play back the human experimental data is made publicly available on GitHub https://github.com/ShortFox/Task-Dynamics-Human-Corralling.

**Funding:** PN was supported by the Macquarie University Research Fellowship. AC was supported by the Charles Phelps Taft Research Center at the University of Cincinnati. MJR was supported by the Australian Research Council Future Fellowship (https://www.arc.gov.au/) (FT180100447). RWK, ES and MJR were supported by the National Institutes of Health (https://www.nih.gov/) (R01GM105045). The funders had no role in study design, data collection and analysis, decision to publish, or preparation of the manuscript.

**Competing interests:** The authors have declared that no competing interests exist.

within these animal-environment systems. These species adopt encirclement strategies whereby members move in coordination to form a dynamic "perimeter wall" to contain and hunt their prey. Specifically, wolves will equally space themselves in a circle around a lone prey to keep the prey immobilized [7], while orcas near Norway will cooperatively encircle herring into a tight ball near the surface in a strategy called "carousel feeding" [8]. Further, humpback whales near both Alaska [9] and Australia [10] have been documented to utilize "bubble-nets" which are produced by blowing bubbles during circular motion below a shoal of fish, giving rise to a cylindrical wall of bubbles that surround and contain the prey.

In addition to what is observed in nature, several types of encirclement strategies have also been documented in laboratory contexts with human participants. In these studies, human dyads engaged in a simulated shepherding game in which participants had to coordinate their hand movements along a tabletop to retrieve and contain a set of evasive target agents (TAs). For this game, participants discovered that an effective solution is to divide the containment perimeter in half, and to produce coordinated *oscillatory* behaviors along their respective half-perimeters to keep the TAs immobilized [11] (see Fig 1). Importantly, this behavior is not adopted by all human dyads, but those who do discover this strategy achieved near-optimal levels of performance. Moreover, in situations where a member of the dyad is required to leave the perimeter in order to retrieve a roaming TA, the participant who remains with the herd readily transitions to producing a continuous *circling* movement around the herd to maintain control [12].

The aim of this paper is to reveal the basis for the similarities in encirclement strategies exhibited by humans in different corralling task contexts. The central premise is that the similarities in the strategies adopted by humans (and other non-human biological or non-biological systems [13]), reflect an alignment of the displayed behaviors to the task's underlying control requirements. The dynamics necessitated by these control requirements can be formalized with a *task-dynamic model* [14–16], which defines the dynamical rules governing the production of movement patterns that ensure a task goal (in this case, containment) is met. Within this framework, the emergence of novel containment behaviors can be understood as being guided by the discovery of latent properties of the underlying task dynamics. This idea was tested by comparing the corralling behaviors adopted by human participants in the current and previous experiments [11, 12, 17] with the emergent behaviors of minimal simulated artificial agents whose dynamics were governed by the task-dynamic model explicitly. Similarities in behaviors in these two systems would provide support for the notion that different

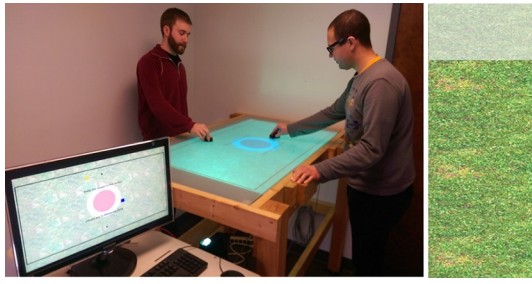
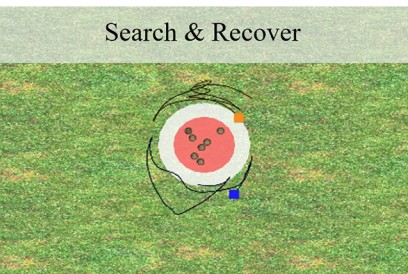
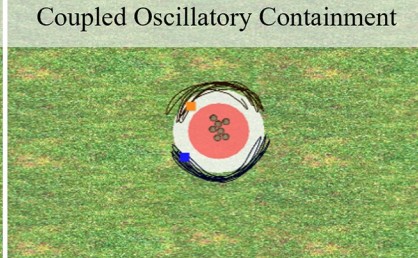

**Fig 1. Experimental setup and behaviors observed in [11].** Participants stood on either side of a projected video display while holding motion sensors which moved their respective herding agents (HAs; the blue and orange square in middle, right panels). At the start of the experiment, participants would sub-divide the task-space to pursue and retrieve target agents (TAs; the brown spheres in middle, right panels) and then keep them contained within the specified circular containment region. Containment was enforced by chasing and retrieving individual escapees back into the region. This strategy is referred to as *search and recover* (S&R) behavior (center panel). Some dyads learned that a far more effective containment strategy was to coordinate oscillatory movements about the TA herd to keep them immobilized. This strategy is referred to as *coupled oscillatory containment* (COC) behavior (right panel). See text for more details.

agents acting in different environment and task contexts exploit similar dynamical rules to achieve task success.

## Background

**Corralling behaviors in humans.** Previous research has utilized corralling task paradigms in a virtual reality framework to explore human dyadic coordination and problem-solving [11, 12, 17, 18]. The task was presented as a video game displayed on a large tabletop display (see Fig 1, left). Standing on opposite sides of the display, participants controlled herding agents (HAs) which would repel nearby TAs. The goal was to keep the TAs from fleeing the game field by containing them within a red circular region during one-minute trials (see [11] for more details, as well as recent implementations [12, 17]). When left unperturbed, the TAs exhibited Brownian motion which required active movements by participants to keep them contained within the red region (else they would easily disperse).

While completing the task, dyads engaged in a behavior termed *search and recover* (S&R) (see Fig 1, center), which involved dyads subdividing the game field (global *'task-space'*) in half and each participant selecting and retrieving the TA farthest from the containment location on their side of the task space. Although this S&R strategy can result in herding all TAs into the containment region, the evasive nature of the autonomous TAs made containing them within this region quite difficult, if not impossible, when the number of TAs increased (e.g., from 3 to 7), with most pairs failing to contain the TA herd using only S&R behavior. Accordingly, some dyads discovered and adopted a much more efficient and effective encirclement strategy to contain the TA herd once the TAs were initially corralled into the containment region using S&R. Instead of each participant pursuing and retrieving individual TAs that escaped the containment perimeter on their respective sides of the task space, both participants would perform coupled, oscillatory movements along their respective half-perimeter (see Fig 1, right). Once discovered, this behavioral containment strategy (*coupled oscillatory containment* [COC]) was immediately implemented by pairs in subsequent trials and led to near-optimal task performance [11]. Moreover, in dyads that discovered the COC strategy, if one participant was left to contain the herd by themselves (e.g., due to the other participant leaving to collect and retrieve a new TA that appeared outside the containment region), that remaining participant would transition to produce circling behaviors around the entire herd [12].

During a post-experiment debriefing interview, dyads who transitioned to COC behavior attributed their discovery to a moment of cognitive insight–i.e., a sudden cognitive reorganization of how to approach the problem [19]. Anecdotally, some participants reported that adopting this solution to solving the task entailed learning to ignore the motions of the contained TAs and, instead, to simply focus on maintaining rhythmic oscillations with their partner. These reports are supported by the stable modes of entrained movements observed in COC-discoverers [11, 17], namely, participants became attracted to producing either in-phase (0°) or anti-phase (180°) coordinated oscillations with their partner. These two coordinative modes of relative phasing are consistent with those displayed across a wide range of bimanual (within-agent) and social (within-dyad) rhythmic coordination tasks [20–24]. Additionally, the discovery and use of COC behavior altered participant eye-movement dynamics, resulting in longer, sustained fixations during COC as opposed to S&R behavior [25]. Collectively, these findings imply that dyads discover a new strategy–namely, to produce coupled rhythmic movements with their partner.

**Task-dynamic models of corralling evasive agents.** The ability to retrieve and contain a set of objects necessitates satisfying task-defined control requirements, which can be formalized using a *task-dynamic model* [14–16]. Rooted in dynamic systems theory, complexity

science and ecological psychology [26, 27], the use of *task dynamics* and its related formulation, *behavioral dynamics* [16], provide a framework for relating intentional individual and collective behavior across animal-environment systems [14, 28]. Task-dynamic models utilize simple mathematical functions or rules (e.g., differential equations) to capture how goal-directed actions by agents unfold at an abstract, low-dimensional level of description [15, 16, 29, 30]. Task-/behavioral-dynamics modeling has previously been applied to human single-agent behaviors such as reaching [15], walking and object avoidance [31], as well as human multiagent activities that entail inter-agent coordination of limb movements [22, 32], sorting and passing objects [33], and crowd motion [6].

To model the corralling behavior of human participants that has been observed in previous studies, the task's dynamics were specified with regard to the goal of minimizing the distance of the set of TAs to a containment region, $\mathbb{C}$. For convenience, the task's dynamics can be defined using a polar coordinate *task-space* $(r,\theta)$, with $(r^{\mathbb{C}},\theta^{\mathbb{C}})$ denoting the center of $\mathbb{C}$ located at the polar origin. The current positions of each $HA_i$ (where $i = 1, 2$ for a human dyad) and their respective TA to pursue at time $t$, $TA_{(t),i}$, can be defined in polar coordinates, respectively, as $(r^{HA_i}, \theta^{HA_i})$ and $(r^{TA_{(t),i}}, \theta^{TA_{(t),i}})$ (see Fig 2). The containment region can be defined in two ways–either as a fixed location on the game field $(x_1^{\mathbb{C}}, x_2^{\mathbb{C}})$, or as a time-varying location centered on the TAs' mean position $(x_1^{\mathbb{C}_{(t)}}, x_2^{\mathbb{C}_{(t)}})$. Once a TA is selected by $HA_i$ at a given time $t$, $HA_i$ begins to retrieve that TA by moving to a task-space location that is slightly radially beyond the task-space location of the selected TA, $(r, \theta) = (r^{TA_{(t),i}} + r_{min}, \theta^{TA_{(t),i}})$. The radial offset, $r_{min}$ (where $r_{min} > 0$), ensures the TA is repelled towards $\mathbb{C}$.

The attraction of a given HA to the location of the TA can be captured using point attractor dynamics. This can be achieved using a damped-mass spring function for both the radial and angular components of the HA's movement,

$$\ddot{r}^{HA_i} + b_r \dot{r}^{HA_i} + \varepsilon_r (r^{HA_i} - (r^{TA_{(t),i}} + r_{min})) = 0 \tag{1}$$

$$\ddot{\theta}^{HA_i} + b_\theta \dot{\theta}^{HA_i} + \varepsilon_\theta (\theta^{HA_i} - \theta^{TA_{(t)i}}) = 0. \tag{2}$$

For Eq 1, $r$, $\dot{r}$ and $\ddot{r}$ represent the radial position, velocity and acceleration, respectively, of $HA_i$ with respect to $\mathbb{C}$; $r^{TA_{(t)i}}$ is the radial position of the TA that $HA_i$ is currently pursuing at time $t$; $r_{min}$ specifies $HA_i$'s preferred radial distance from the selected TA; and $b_r$ and $\varepsilon_r$ are free parameters that vary the damping and stiffness, respectively, of the radial force attracting $HA_i$ from $r^{HA_i}$ to $(r^{TA_{(t)i}} + r_{min})$. Eq 2 mirrors Eq 1 but controls the angular force attracting $\theta^{HA_i}$ towards $\theta^{TA_{(t)i}}$.

When faced with multiple TAs to select from, a HA is hypothesized to implement the intuitive rule of targeting and moving towards the TA that is (i) farthest from $\mathbb{C}$ and (ii) moving away from $\mathbb{C}$ [11, 12, 17]. In the case of multiple HAs, each HA implements the rule of selecting the farthest TA that belongs to a subset of TAs closer to one's own position compared to their partner's. These TA selection rules can be formalized as the following,

$$\{TA_{(t),i} \in H_i | \max (\|\boldsymbol{x}^{TA_{(t)i}} + \dot{\boldsymbol{x}}^{TA_{(t)i}} \Delta t\|)\}. \tag{3}$$

For Eq 3, the TA to pursue at time $t$, $TA_{(t),i}$, is a member of the subset of TAs, $H_i$, which are closer to $HA_i$ than their partner (see **Method** for more detail). From this subset, $TA_{(t),i}$ is the TA who will be farthest from $\mathbb{C}$, where $\boldsymbol{x}^{TA_{(t)i}}$ and $\dot{\boldsymbol{x}}^{TA_{(t)i}}$ represents the position and velocity vectors, respectively, of $TA_{(t),i}$ in relationship to $\mathbb{C}$ in Cartesian space. The value $\dot{\boldsymbol{x}}^{TA_{(t)i}} \Delta t$ represents the positional increment added to $\boldsymbol{x}^{TA_{(t)i}}$ at time $t$ to predict $TA_{(t),i}$'s position at $t+\Delta t$. When incorporating a TA selection rule such as Eq 3, the task-dynamic model defined in Eqs

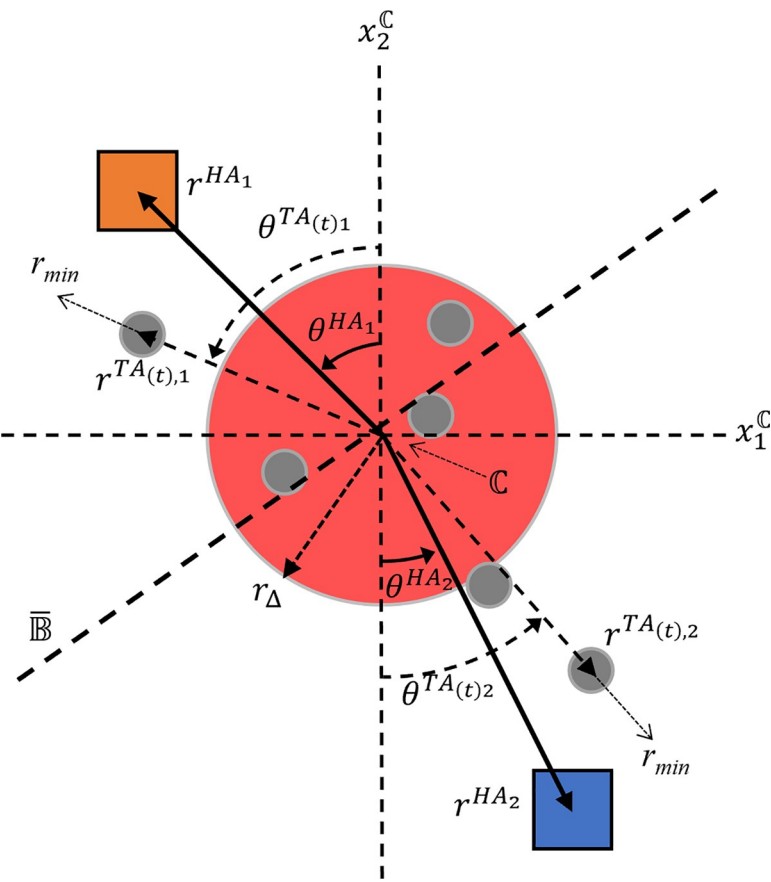

**Fig 2. Pictorial representation of the task-space for human herding and containment.** The position of the $i^{th}$ herding agent (HA$_i$; where $i$ = 1 [orange square], 2 [blue square]) is defined in a polar coordinate task-space ($r^{HA_i}$, $\theta^{HA_i}$) with the center of the containment region (the red circle), $\mathbb{C}$, defined at the polar origin. At time $t$, each HA$_i$ moves towards the position of the TA, ($r^{TA_{(t),i}}$, $\theta^{TA_{(t)i}}$), that is both $i$) a member of the subset of TAs closer to itself than to its HA partner and $ii$) is predicted to be the farthest TA in this subset from the center of the containment region at time at time $t+\Delta t$. This subset can be visualized using the boundary $\bar{\mathbb{B}}$, defined as the perpendicular bisector of a line drawn between the HAs; the TAs falling on HA$_i$'s side of $\bar{\mathbb{B}}$ comprise the set of potential targets for HA$_i$. Parameter $r_{min}$ represent an offset to ensure TAs are repelled towards $\mathbb{C}$, and $r_\Delta$ represent a decision boundary that determines the minimum distance of the selected TA that elicits herding behaviors by an HA. See text for more details.

1 and 2 is sufficient to generate S&R behavior. However, the dynamics defined by Eqs 1 and 2 cannot produce either the coupled oscillatory containment (COC) behaviors adopted by human dyads [11, 17] or the circling behaviors observed by individual human participants [12]. To account for COC behavior by dyads, the angular dynamics defined by Eq 2 can be modified to capture both the oscillatory nature of COC, as well as the stable patterns of inter-participant entrainment. This can be done by converting the damped mass-spring dynamics to limit-cycle oscillatory dynamics, and by adding an appropriate inter-HA coupling function,

$$\ddot{\theta}^{HA_i} + b_\theta \dot{\theta}^{HA_i} + \beta_\theta \dot{\theta}^{3\ HA_i} + \gamma_\theta \theta^{2\ HA_i} \dot{\theta}^{HA_i} + \varepsilon_\theta(\theta^{HA_i} - \theta^{TA_{(t)i}})$$
$$= (\dot{\theta}^{HA_i} - \dot{\theta}^{HA_j})(A + B(\theta^{HA_i} - \theta^{HA_j})^2) \tag{4}$$

The inclusion of the nonlinear terms $\beta_\theta \dot{\theta}^{3\ HA_i} + \gamma_\theta \theta^{2\ HA_i} \dot{\theta}^{HA_i}$ enables oscillatory, limit-cycle dynamics when $b_\theta < 0$. The coupling term $(\dot{\theta}^{HA_i} - \dot{\theta}^{HA_j})(A + B(\theta^{HA_i} - \theta^{HA_j})^2)$, when $|4B| > |A|$, couples the angular dynamics of HA$_i$ to its partner HA$_j$ to produce the stable in-phase or anti-

phase coordinative patterns observed in previous research [17] and in rhythmic coordination more generally [20, 21, 34]. A more detailed discussion of Eq 4 can be found elsewhere [17, 35].

In the case of an individual HA producing circling movements to keep the TA herd contained, Eq 2 can be modified as follows,

$$\ddot{\theta}^{HA_i} - b_\theta \dot{\theta}^{HA_i} + \beta_\theta \dot{\theta}^{3\,HA_i} = 0. \tag{5}$$

When $b_\theta < 0$, an HA implementing Eq 5 will circle around the task-space origin at a rate of $\pm\sqrt{\frac{b_\theta}{\beta_\theta}} rad \cdot s^{-1}$ with the direction specified by the HA's initial angular velocity [15]. Further, to accommodate the observation that participants will perform COC or circling behaviors at a fixed distance from the task-space origin, Eq 1 can also be modified to replace $(r^{TA_{(t),i}} + r_{min})$ with $r_\Delta$, where $r_\Delta$ represents an HA's preferred distance from $\mathbb{C}$. Both Eqs 4 and 5 have been successfully implemented into the control architecture of an artificial HA capable of working alongside naïve participants to contain TAs to a singular location [17], as well as transporting between multiple locations and adapting to the introduction of new TAs during a trial [12].

**Current study.** The transition from S&R behavior to containment and circling behaviors is driven by a change in how HAs are coupled to the task environment. In a task-dynamic framework, these changes in behavior are induced by corresponding *graph-dynamic* [30] changes in the compositional structure of the system equations that underlie the observed behavior (Eqs 1 and 2 for S&R, and Eqs 4 and 5 for containment and circling behavior, respectively). The discovery of containment strategies by participants in previous research may reflect context-dependent interactions that facilitate the detection of these behavioral possibilities. In this way, Eqs 4 and 5 may exist as latent properties revealed by the movement patterns shaped by Eqs 1 and 2. Accordingly, when engaged in a particular corralling task context, human actors are hypothesized to exploit these latent properties and subsequently implement these dynamics intentionally [28, 36].

Inspired by the similitude in the containment and encirclement strategies observed during human [11, 12, 17], as well as animal [7–10] and non-biological systems [13], the current study evaluated whether COC and circling behaviors can be understood more generally as invariant, emergent properties of human dyadic corralling behaviors. As opposed to limiting the behaviors of participants to hand movements on a tabletop display, participants were embodied in an immersive virtual reality environment in which they had to locomote across a large space to corral and contain the fleeing TA herd (see Fig 3). Model-based simulations were also conducted to determine whether the emergent behaviors that participants adopt in this current experiment, as well as in previous research [11, 12, 17], can be predicted as latent properties of the task-dynamic model.

# Results

Human naïve participants, recruited as dyads, were tasked to keep a set of seven TAs (modeled as spheres with radius 0.24 m) contained within 0.72 m of the TA herd's mean position on a game field measuring $6 \times 3.48$ m (see Fig 3). Dyads were exposed to one of three difficulty conditions, which manipulated the maximum speed the TAs could move ($\leq 0.12$ m·s$^{-1}$, $\leq 0.20$ m·s$^{-1}$, $\leq 0.28$ m·s$^{-1}$). Dyads were given 45 minutes to solve the task across a series of two-minute trials. During a trial, dyads had to prevent any TA from fleeing the field. Additionally, participants had to contain the TAs within the containment criteria for at least 70% of the last 45 seconds of the trial. Dyads who could meet these criteria on eight separate trials were deemed successful.

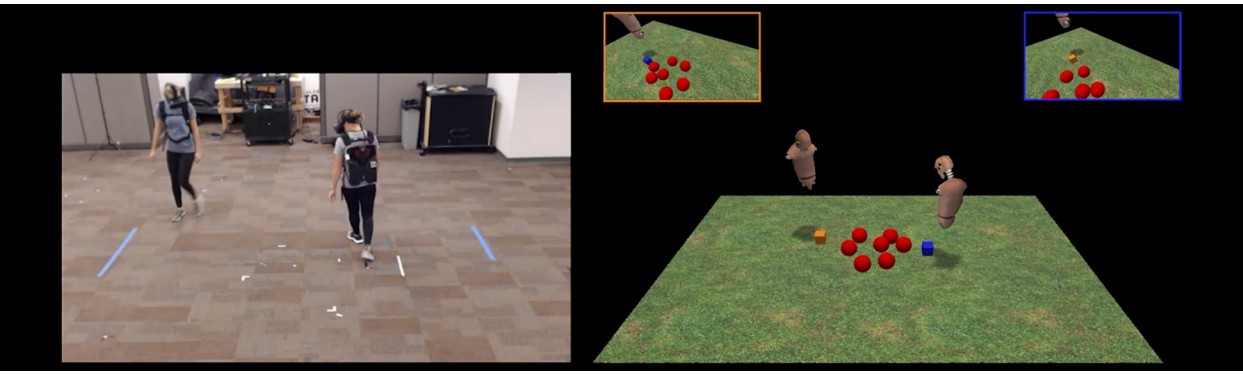

**Fig 3. Depiction of experimental setup and task environment.** Participants (left) wore virtual reality headsets connected to a computer worn as a backpack. Participants embodied floating avatars (right) and could locomote about a large (6 × 3.48 m) environment to corral and contain seven evasive target agents (TAs; red spheres) within 0.72 m of the TA herd's mean position. Task difficulty was manipulated by clamping the maximal speed of the TAs' movements ($\leq$ 0.12 m·s$^{-1}$, $\leq$ 0.20 m·s$^{-1}$, $\leq$ 0.28 m·s$^{-1}$). The TAs were reactive to the orange-/blue-colored cubes at the base of each participant's avatar, which was controlled via their head movements. The orange- and blue-bordered images show the participants' respective perspectives in their virtual reality headsets. See **S1–S3 Videos**.

In total, 39 dyads participated in the experiment. These dyads were equally distributed across low-($\leq$ 0.12 m·s$^{-1}$), medium-($\leq$ 0.20 m·s$^{-1}$) and high-($\leq$ 0.28 m·s$^{-1}$) TA maximal speed conditions. From the 39 dyads, 26 (66.67%) dyads met the experiment success criteria. Most failures resulted from dyads who were assigned to the high-speed condition (3 [23.08%] dyads reached eight successful trials), followed by the medium-speed condition (10 [76.92%] dyads), whereas all dyads in the low-speed condition succeeded. Within these successful dyads, the time required to complete the experiment was 16.91 ($SD$ = 1.23) minutes, 24.83 ($SD$ = 4.73) minutes, and 32.02 ($SD$ = 4.73) minutes for the low-, medium- and high-speed condition, respectively. Overall, the range of task difficulty was sufficiently broad so that the low-speed condition was trivially easy, and the high-speed condition was very difficult.

## Humans discovered coordinated circling as an effective corralling strategy that was sensitive to task difficulty

Across conditions, successful dyads discovered that an effective means to complete the task was to perform circling movements around the TA herd. Representative examples of these movement patterns and how they varied across task difficulty conditions are shown in Fig 4. Across successful containment trials, dyads performed 5.99 ($SD$ = 4.04), 7.62 ($SD$ = 4.24), and 13.15 ($SD$ = 4.81) *cumulative 2π rotations* in the low-, medium-, and high-speed difficulty conditions, $F(2, 31)$ = 7.12, $p$ = .003, $\eta_p^2$ = .32. The number of rotations completed by dyads in the high-speed condition was greater than those in the low- ($p$ = .002) and medium-speed ($p$ = .02) conditions, where pairwise comparisons for this and all other tests were Bonferroni-corrected. As shown in Fig 4, dyads in the lower speed conditions often exhibited intermittent cycling or cycling with frequent direction changes. This was in contrast with dyads in the high-speed condition who maintained a fixed rotation direction over the course of the entire trial. Despite these differences in locomotion cyclicity, participants remained coordinated in their behaviors by maintaining an 180˚ (π radians) *angle of separation* with respect to the TA's mean position, $t(33)$ = -0.31, $p$ = .76, which did not differ between conditions, $F(2, 31)$ = 0.99, $p$ = .38, $\eta_p^2$ = .06. Additionally, *dyad cycling direction* preference was symmetrical–a particular dyad was equally likely to prefer a clockwise (CW) or counterclockwise (CCW) rotation around the TA herd (CCW rotation during 56.75% of successful trials, $SD$ = 32.26), $t(33)$ = 1.22, $p$ = .23, $d$ = 0.21).

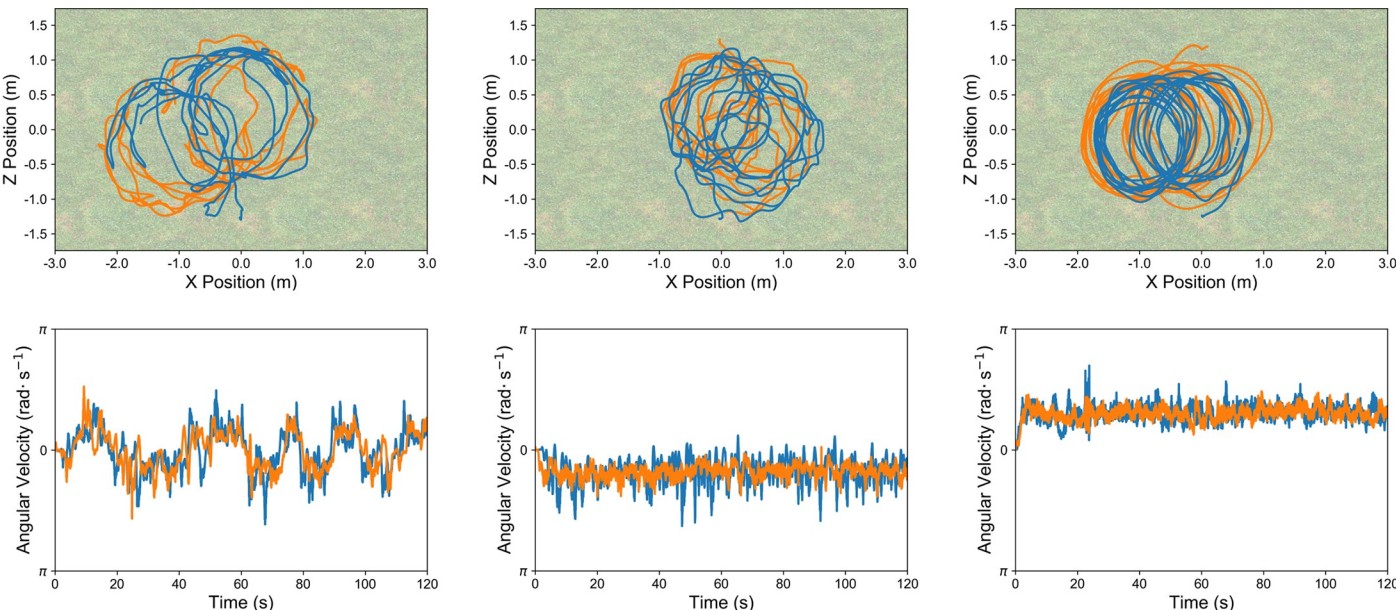

**Fig 4. Representative examples of intermittent clockwise (CW)/counterclockwise (CCW) (left, from low-speed condition), continuous CW (center, from medium-speed condition) and continuous CCW (right, from high-speed condition) circling behavior observed in the experiment with human participants.** The top row shows the paths taken by participants and the bottom row shows their corresponding angular velocity time series of both participants with respect to the TAs' mean position at each timestep. Regardless of the mode of behavior, participants' movements remained coordinated with each other, maintaining an angular difference of ≈180° with respect to the target agents' (TAs') mean position. See **S1–S3 Videos** for the accompanying videos of the left, center, and right panels, respectively.

Although most dyads in the high-speed condition did not reach the success criteria for the experiment, when considering all successful trials obtained from these dyads ($M = 6.25$ trials, from 8 dyads), containment performance was similar (*TA containment time* $M = 111.83$ s, $SD = 7.12$) to those in the low-speed ($M = 117.72$ s, $SD = 2.17$) ($p = .22$) and medium-speed conditions ($M = 104.95$ s, $SD = 9.69$) ($p = .11$), $F(2, 31) = 10.69$, $p < .001$, $\eta_p^2 = .41$. Interestingly, performance was worse on successful trials from dyads in the medium-speed condition as compared to those in the low-speed condition ($p < .001$). This decrease in performance may reflect an approach towards a critical point that destabilizes intermittent (CW/CCW) circling behavior as an effective solution to sufficiently control the TAs' movements. Although there was no difference in the number of rotations performed by dyads in the low- ($M = 5.99$ rotations, $SD = 4.04$) and medium-speed ($M = 7.62$, $SD = 4.24$) conditions ($p > .99$), dyads in the medium-speed condition did not restrict the *TAs' distance travelled* ($M = 11.52$ m, $SD = 2.14$) as much as dyads in the low- ($M = 8.89$ m, $SD = 0.71$) ($p = .001$) and high-speed (9.05 m, $SD = 1.74$) ($p = .006$) conditions, $F(2, 31) = 9.96$, $p < .001$, $\eta_p^2 = .39$. As the maximal speed of the TAs increased, a transition from intermittent to continuous cycling in a fixed direction appeared necessary to maintain control over the herd. Dyads in the high-speed condition managed to restrict the TAs' movements to the same degree as those in the low-speed condition ($p > .99$).

## Model simulations revealed emergent dynamics that emulated the strategies humans adopt in different corralling task contexts

The cyclical behaviors human dyads produced while locomoting are different from the oscillatory behavior participants discover when using hand movements to control their respective

HA (as opposed to the head's position used in the current experiment). A notable difference between the tabletop environment (Fig 1) and the environment employed here (Fig 3) is the effort required to traverse and change directions in the task fields for hand movements (field width = 1.17) [11, 17] and locomotion (field width = 6 m). Model-based simulations were conducted to determine whether both circling and oscillatory behaviors could be understood as latent, emergent properties of the same task-dynamic model when appropriately parameterized to account for the differential constraints acting upon HAs in either context. In this way, the structure of the task-dynamic model reflects the constraints of the task while the model's parameterization reflects the physical constraints acting upon HAs to move about the task environment [37]. Specifically, the dynamics afforded by Eqs 1 and 2 can be constrained by the parameterization by the model's stiffness, $\varepsilon$, and damping, $b$, parameters, which relate to the conservative and dissipative forces acting upon the system, respectively.

Two artificial HAs completed the corralling task in the task environment used for the human experiment. The agents embodied the model defined by Eqs 1, 2 and the TA selection rule (Eq 3). Simulations were conducted across a range of stiffness and damping parameter values. The same parameter values were used for the radial and angular dynamics (i.e., $\varepsilon = \varepsilon_r = \varepsilon_\theta$; $b = b_r = b_\theta$). Plots summarizing the simulations are presented in Figs 5 and 6.

When underdamped ($\zeta<1$), emergent oscillatory behavior can be observed in Fig 5 (bottom row). This oscillatory behavior was the result of a symmetry-breaking event once the TAs were contained, whereby the repulsions of the artificial HAs caused the TAs to move collectively in the same direction. This resulted in a stable pattern of behavior whereby the artificial HAs "oscillated" to negate the directed forces of the TA herd (see **S4 Video**). Note that these COC-like movements were reactionary to the movements of the TAs and were not internally generated.

Although this emulated the behaviors observed when participants complete the corralling task using hand movements on a tabletop display [11, 17], this behavior was not tenable when locomoting a large task environment. Specifically, when considering parameters that resulted in a *mean peak frequency* > 0.5 Hz, the cutoff criterion for COC behavior [11, 17], the movements of the artificial HAs equated to approximately 2.67 m·s$^{-1}$ ($SD = 0.73$), exceeding the transition threshold from walking to running in humans (~2.0 m·s$^{-1}$) [38]. Given this behavior includes the reversals inherent in COC behavior, this speed is not maintainable for a substantial period (e.g., two-minutes).

However, as the movement speed of the artificial HAs decrease (i.e., due to overdamped dynamics, $\zeta>1$), a different symmetry-breaking event occurred (Fig 5, top half). As opposed to exhibiting oscillatory-like behavior, the artificial HAs exhibited emergent circling behavior around the TA herd while maintaining an *angle of separation* of ~180˚ from each other. This behavior resulted from the artificial HAs' inability to move to the selected TAs fast enough to direct a repulsion force towards the TAs' mean position. Instead, an oblique force is applied, inducing a collective spin in the TAs (see **S5 Video**) which the HAs reactively follow. When considering parameters resulting in a *mean peak frequency* of ≤ 0.5 Hz, the corresponding *HA movement speed* was approximately 0.91 m·s$^{-1}$ ($SD = 0.46$), which reflects the preferred walking speeds of participants during circular turning– 0.96 ± 0.1 m·s$^{-1}$ [39]. Given these settings, the HAs on average produced 4.67 *cumulative 2π rotations* around the TA herd ($SD = 1.93$), as opposed to 0.91 *cumulative 2π rotations* ($SD = 0.46$) when the agents' parameterization resulted in a *mean peak frequency* of > 0.5 Hz. Note, rotations when *mean peak frequency* was > 0.5 Hz were due to transient behaviors by the artificial HAs at the beginning of the simulation. Once the TAs were contained, the artificial HAs stabilized on in-phase oscillations (see Fig 5, bottom left).

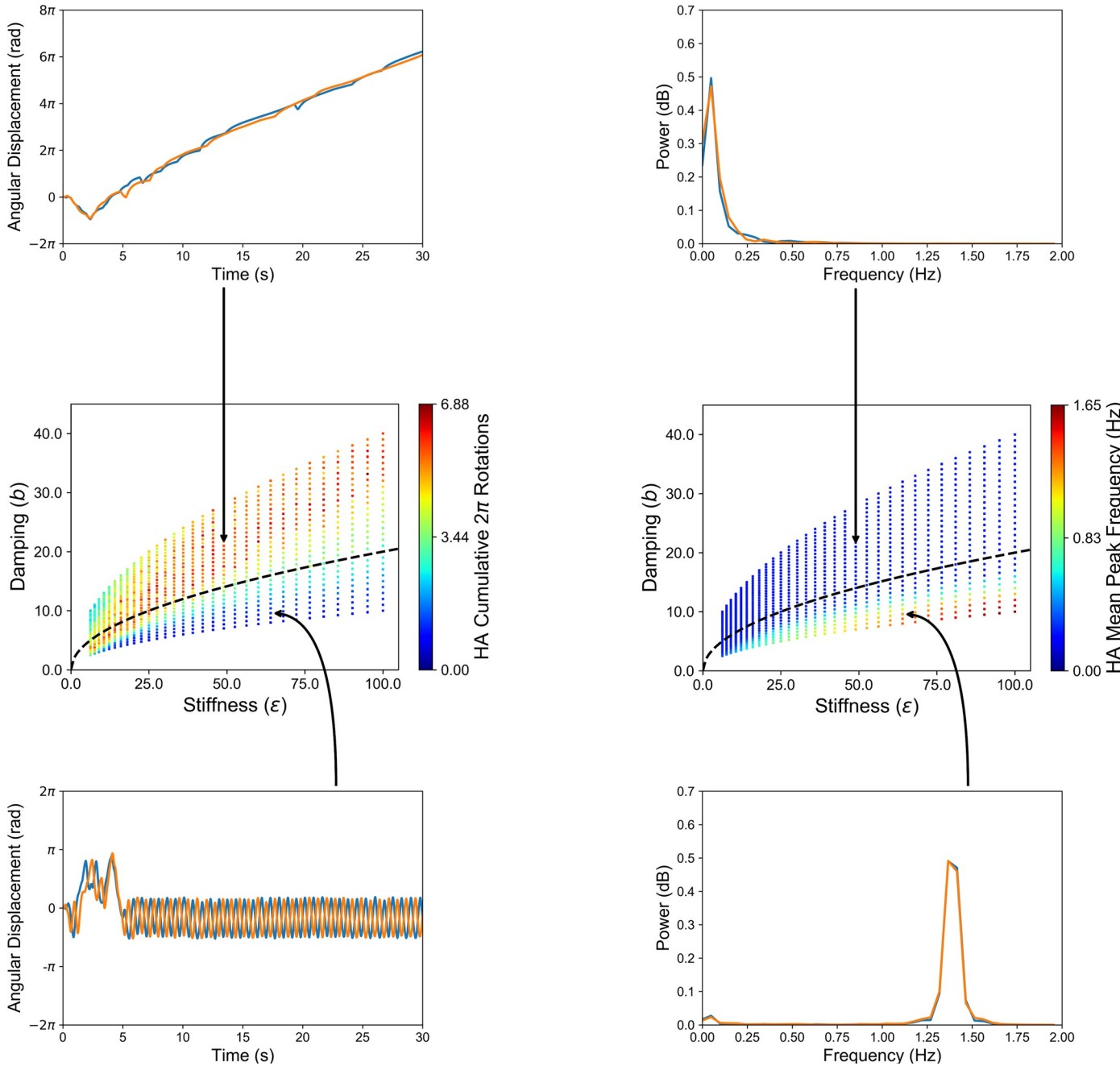

**Fig 5. Two different emergent behaviors as a function of changes to stiffness and damping (middle row), as illustrated by the *cumulative 2π rotations* (middle left) and the *mean peak frequency* of angular movement (middle right) by artificial herding agents (HAs).** The black dashed line represents critical damping ($\zeta = 1$), which satisfies the condition $b = 2\sqrt{m\varepsilon}$ ($m = 1$ kg). Parameters above the line indicate overdamping ($\zeta > 1$), while those below indicate underdamping ($\zeta < 1$). Artificial HAs embodying the task-dynamic model defined in Eqs 1–3 exhibited circling movements around the target agent (TA) herd when the model was overdamped (top left). When the model was underdamped, the two artificial HAs exhibited oscillatory behavior (bottom left) at a peak frequency consistent with previous work studying human collaborative problem-solving (bottom-right power spectrum) [11, 17]. Angular displacements in the negative/positive direction indicate counterclockwise/clockwise motion. See also **S4 and S5 Videos**.

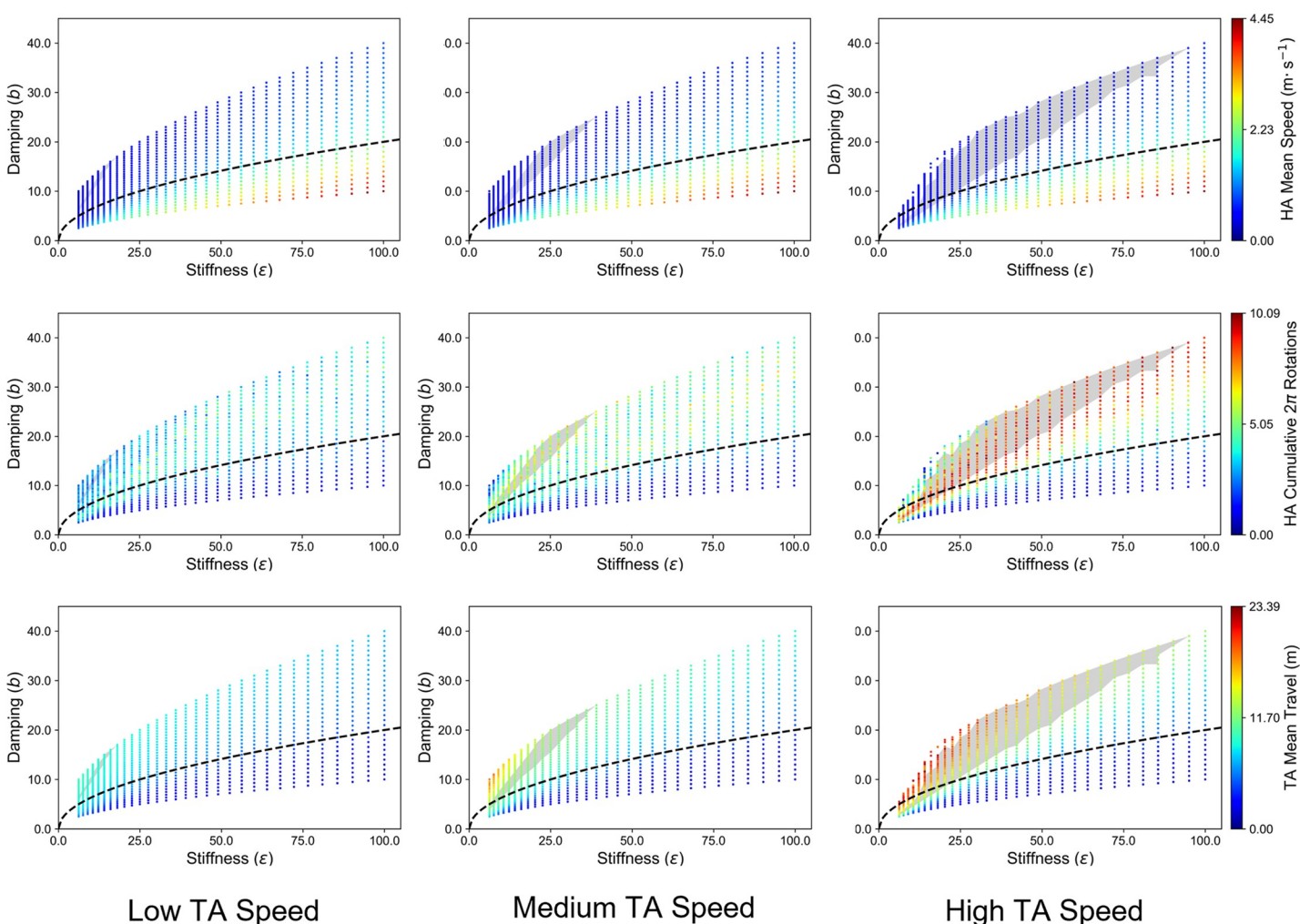

Low TA Speed    Medium TA Speed    High TA Speed

**Fig 6. Relationship between human and artificial herding agent (HA) behavior at different task difficulty conditions.** The colored scatterplots on all nine figures represent the data points associated with the combinations of stiffness ($\varepsilon$) and damping ($b$) values used in the HA simulations (see main text). The black dotted line running through each of these figures represents critical damping, i.e., those parameter values that satisfy the condition where $b = 2\sqrt{m\varepsilon}$, and $m$ was set equal to a constant value of 1 kg. Parameters above/below the line indicate a system that is over/under-damped. The color coding of data points is defined according to *mean movement speed* of the HAs (top row), number of *cumulative 2π rotations* by the HAs (middle row), and *mean distance traveled* (i.e., path length) by the target agents (TAs; bottom row). From left to right, figure columns represent simulations at three increasing levels of task difficulty according to correspondingly increasing levels of assigned TA speeds: Low-speed ($\leq 0.12$ m·s$^{-1}$; left column), medium-speed ($\leq 0.20$ m·s$^{-1}$; middle column), and high-speed ($\leq 0.28$ m·s$^{-1}$; right column). Different combinations of stiffness and damping resulted in differences in the qualitative behaviors adopted by the artificial HAs (middle row), as well as in the HAs' ability to contain the TAs (bottom row). The gray areas in all figures designate the regions of $b$-$\varepsilon$ parameter space that fall within the 95% confidence interval of human locomotion speeds in these difficulty condition (constructed using HA Mean Speed [top row]). See also **S1 Fig**.

## Model simulations reproduced the effect of task difficulty on human coordinated circling behavior

The role of task difficulty in the emergence of circling behavior exhibited by both human participants and artificial HAs becomes clearer when comparisons are made between human participant and artificial HA behavior, with model simulation parameters selected which result in movement performance in the range of the human dataset. As shown in Fig 6, combinations of stiffness and damping are plotted which resulted in artificial *HA movement speeds* that fell within the 95% confidence interval of human participant locomotion speed (low-speed: [0.39, 0.45] m·s$^{-1}$ 95% CI; medium-speed: [0.45, 0.55] m·s$^{-1}$ 95% CI; high-speed: [0.58, 0.72] m·s$^{-1}$

95% CI) (the gray areas in Fig 6). When constrained to reflect the movement performance of human participants, the simulations reproduced the qualitative behaviors observed in the current experiment. Specifically, as task difficulty increased, the artificial HAs produced more cyclical behaviors in a fixed direction (low-speed $M = 3.02$ *cumulative 2π rotations*, $SD = 1.02$; medium-speed $M = 3.93$, $SD = 1.52$; high-speed $M = 5.60$, $SD = 2.74$). However, unlike the results from human participants, as task difficulty increased, so did the *TAs' distance travelled* (low-speed $M = 6.54$ m, $SD = 2.25$; medium-speed $M = 8.47$, $SD = 3.79$; high-speed $M = 10.77$, $SD = 5.77$), highlighting that the cyclical behaviors of the artificial HAs was in response to the fleeing behavior of the TAs.

## Discussion

Novice participants, when tasked to corral seven evasive target agents (TAs) in a task environment which required locomotion, developed a coordinated circling strategy which kept the agents sufficiently contained. This circling behavior was distinct from the oscillatory (i.e., COC) behavior observed by dyads in previous research where participants completed the task using hand movements [11, 17]. However, simulations demonstrated that both behaviors can be understood as emergent properties of the same underlying task dynamics, captured using the task-dynamic model detailed in Eqs 1, 2 and the TA selection rule (Eq 3). Further, this model can also reproduce individual circling behavior as witnessed in previous research (see **S6 Video**) [11, 12].

The task-dynamic model presented here shares features with other bio-inspired modeling approaches. Strömbom et al. [40] modeled Australian sheepdog retrieval behavior when pursuing sheep that are farthest from the herd and driving the sheep to the herd's center of mass. Further, Muro et al. [7] demonstrated that various complex group hunting behaviors in wolves can be recreated by two simple rules followed by individual wolves–namely, to move directly towards the prey until a safe distance is reached, and then to move away from neighboring wolves during containment. There are notable differences, however, between the corralling task that human participants completed in the present study and what is observed in nature. For instance, the TAs corralled by participants in this study, as well as in previous work [11, 12, 17], exhibited dynamics different from what is observed in natural herding and hunting contexts. Unlike real animals who clump when threatened [41], the TAs in this study were not coupled to each other. Thus, participants in this study had to continually act to keep the herd contained by keeping the forces directed towards the herd's center of mass. In contrast, canids either make periodic movements to contain fleeing agents escaping from the flanks of the herd, as is the case in shepherding [40], or equally disperse themselves around a prey, as is the case in wolf-pack hunting [7]. However, when the escape capabilities of prey increase, more explicit encirclement behaviors are observed. For example, cetaceans like orcas [8] and humpback whales [9, 10] exhibit carousel/bubble-net feeding whereby members of a pod will encircle and blow bubble ring structures to entrap fish. In non-animal systems where the artificial herders' task is to corral large numbers of artificial agents that do not have preferences to clump, explicit circling behaviors are also adopted [13]. Thus, due to the relative difficulty of the task, circling behaviors by human participants reflected a coordination strategy that maximized control to keep the TA herd contained.

The discovery of encirclement behaviors by participants may reflect interaction dynamics that scaffold their realization [42]. As demonstrated through simulation, artificial agents implementing the task-dynamic model defined in Eqs 1–3 unveiled oscillatory and circling behaviors as a latent, emergent property of the task context. Similarly, what may differentiate high performing from unsuccessful human dyads are differences in interactions which govern

whether these latent dynamics are generated. Or, once generated, participants may vary in their ability to detect these latent possibilities to guide interpersonal coordination. Once detected, however, dyads can then learn to exploit this emergent property intentionally [28, 36], removing the dynamic as a latent feature of the task, and instead producing these dynamics explicitly.

The discovery and utilization of explicit coordinated circling by participants in this experiment can be modeled by modifying Eq 5, which defines individual circling behavior, to include a repulsive reactive coupling term, $-(\theta^{HA_i} - \theta^{HA_j})$. This term serves to maximally separate $HA_i$ from its co-actor $j$ while circling, resulting in an angular difference of 180°, consistent with what was observed in the human experiment,

$$\ddot{\theta}^{HA_i} - b_\theta \dot{\theta}^{HA_i} + \beta_\theta \dot{\theta}^{3\ HA_i} = -(\theta^{HA_i} - \theta^{HA_j}). \tag{6}$$

Given the different corralling behaviors that can be modeled by Eqs 2 and 6, respectively, a step function can be used to switch between either equation,

$$\ddot{\theta}_{HA,i} = \begin{cases} -b_\theta \dot{\theta}^{HA_i} - \varepsilon_\theta(\theta^{HA_i} - \theta^{TA(t)i}), & if\ r^{TA(t),i>r_\Delta} \\ b_\theta \dot{\theta}^{HA_i} + \beta_\theta \dot{\theta}^{3,HA_i} - (\theta^{HA_i} - \theta^{HA_j}), & if\ r^{TA}(t),i \le r'_\Delta \end{cases}, \tag{7}$$

whereby Eq 2 is implemented if the pursued TA's radial position, $r^{TA(t),i}$, exceeds the distance of the containment threshold, $r_\Delta$ (see Fig 2), otherwise Eq 6 is implemented. Thus, in addition to discovering the latent dynamics defined by Eq 6, participants are also hypothesized to learn to detect contextual information which determines the appropriateness of circling behavior via a control law [16] (e.g., Eq 7).

In skill learning, an individual develops coordination patterns, or synergies [43, 44], which define how the many degrees-of-freedom of one's body should interact to allow for the control of movement [43]. Once these synergies are formed, they must be appropriately coupled to environmental information that specifies the control requirements for a given task, which are low-dimensional in comparison. This interactive coupling between agents and their environment can be formalized using the task-/behavioral-dynamic modeling framework discussed in this paper [45]. The challenge when moving from individual to collective and collaborative task contexts is to understand how groups divide a task between members. An answer to this problem requires both an understanding of what information agents use to make individual decisions in group contexts [46, 47], as well as how these decisions are communicated to others agents through coupling [48–52]. As with individual behavior, human groups can form *interpersonal* synergies whereby individuals in a collective will provide compensatory support along task-relevant dimensions to help achieve a joint goal [53, 54]. This ability to coordinate is maximal when agents are similar in their movement kinematics or complexity [55–57].

However, the identification of the task-relevant components that need to be controlled may not always be transparent. Recent advancements in machine learning approaches may provide an opportunity, in conjunction with theoretical modeling, to uncover the necessary control laws and coordination couplings [58]. Agents trained using deep reinforcement learning (DRL; i.e., the integration of reinforcement learning with deep neural networks), for example, have been successful in discovering adaptive behavior and strategies in individual [59] and group task contexts [60, 61]. Within the context of working with humans in collaborative tasks, such agents can develop control policies that are either user-specific [62] or generalize to a distribution of human strategies during training [63]. By giving meaning to actions with the use of reward functions [64], black-box self-supervised approaches have the ability to provide a "direct fit" [65] between an agent and task-relevant states–assuming there is sufficient

sampling of the task environment. Indeed, the reason why deep neural networks may be so successful in certain task domains may be because of their ability to detect the low-dimensional structure of the world [66]. As demonstrated with the recent successes in DRL, deep neural network architectures can not only detect invariances in physical properties such as in image detection [67], but may also be able to detect the low-dimensional structure of animal-relevant properties (i.e., affordances [26]), which can be formalized using task-dynamic models, in constraining animal, including human, behavior.

## Method

### Participants

Eighty undergraduate students from the University of Cincinnati (*M* age = 18.93 years, *SD* = 0.90), recruited as dyads, participated in the experiment. One dyad was later removed from analysis due to a premature software closure. Participants received research credit towards completion of a Psychology course requirement. The study was approved by the University of Cincinnati's Institutional Review Board.

### Materials and task

The corralling task was designed as an immersive virtual reality experience where dyads could locomote in a shared physical and virtual space to retrieve and contain evasive target agents (TAs) see Fig 3). The task software was developed using Unity (ver. 5.6.12, Unity Technologies, San Francisco, USA) and a wireless local area network (WLAN) was utilized to synchronize task and participant states using Unity's UNET server-authoritative networking protocol. Participants wore backpacks containing portable computers (MSI VR-One, Micro-Star International, Taiwan) which were equipped with an HTC Vive virtual reality headset (HTC Inc., Taiwan). This setup allowed for the free movement of participants within an $8 \times 6$ m space. Within the virtual environment, participants were embodied as floating avatars (avatar shoulder width = 0.42 m), calibrated to each participant's height. The avatars were controlled by the participants' head movement and an inverse kinematics controller (FinalIK, Rootmotion, Estonia) was responsible for rotations of the avatar's torso. Task and participant states were recorded at 50 Hz.

 The virtual environment contained a grass field ($6 \times 3.48$ m), seven TAs, and a herding agent (HA) for each participant. The TAs were modeled as spheres (radius = 0.24 m, mass = 2 kg) and their native behavior was governed by Brownian motion. The TAs could leave the task space by falling over the edge of the grass field. Each participant's HA was represented as a blue or orange colored cube (edge length = 0.15 m) with the cubes' positions ($x$, $z$) set to their respective participant's head position at each timestep. Interactions with the TAs by participants were done through their HA cubes. Whenever a participant's HA was within 0.6 m of a TA, the Brownian force acting upon the TA was replaced by a repulsive force directed away from the participant's HA (the dynamics of which is consistent with the method reported in [11]). The maximum speed the TAs could move was clamped to not exceed the value set by the task's condition ($\leq 0.12$, $\leq 0.20$ or $\leq 0.28$ m·s$^{-1}$). This repulsion force was the only means of interacting with the TAs. Collisions between HAs and TAs were not possible, although TAs were able to collide with each other.

 Before each trial, participants moved their respective HA to a start location ($x$, $z$) = (0, ±1.2 m) on their respective side to jointly initiate a trial. Once initiated, seven TAs appeared centrally clustered on the grass field between both participants (cluster center [$x$, $z$] = [0, 0]; cluster radius = 0.36 m). The aim of the task was to contain all seven TAs within 0.72 m of the TA's mean position (i.e., the TAs' centroid), calculated at each timestep. When all TAs were within

this threshold, the color of the TAs turned red (otherwise they remained a white color). Each trial was 2 minutes in length, and a trial was completed successfully if all TAs were sufficiently contained for at least 70% of the last 90 s of the trial (i.e., for at least 63 seconds). If the 70% criterion was not met, or if a trial ended prematurely due to a TA falling over the edge of the grass field, the trial was considered a failure. At the end of each trial, participants received visual feedback regarding their performance. If the entire 120 s trial duration elapsed, the feedback was the percentage of time the TAs were contained during the last 90 s of a trial. If the trial ended prematurely, the visual feedback was a message that read "Try Again!".

## Design and measures

The experiment with human participants implemented a between-subject design. Dyads were randomly assigned to one of the three maximal TA speed conditions (low-speed = 0.12 m·s⁻¹; medium-speed = 0.20 m·s⁻¹, high-speed = 0.28 m·s⁻¹). Task performance as well as human participant and simulated agent behaviors were assessed for successful trials. Measures were calculated for the entire trial duration (120 s). The measures used to assess task performance were *TA containment time* (s)–the total time the TAs were within the containment criteria–and *TA distance travelled* (m)–the mean TA path length (cumulative sum of the displacements).

The following measures were used to assess both participant and simulated (see Model simulations) HA behavior. First, the degree to which HAs performed cyclical movements around the TA herd was computed by taking the cumulative sum of the angular change of their movement with respect to the TA herd's mean position, divided by $2\pi$ (referred to as *cumulative $2\pi$ rotations*). The absolute value of the result was taken to account for preferences for clockwise/counterclockwise motion. Relatedly, *dyad cycling direction* was the sign of *cumulative $2\pi$ rotations* prior to taking the absolute value. A negative/positive value indicated a preference for clockwise/counterclockwise motion. Next, to quantify the coordination between HAs, the *angle of separation* was computed as the mean angle between $x^{HA_1}$ to $x^{HA_2}$ in the counterclockwise direction (i.e., from 0 to $2\pi$), where $x$ is the position vector with respect to the TA herd's mean position. Finally, *HA movement speed* (m·s⁻¹) was defined by calculating the mean cumulative sum of the displacements by both HAs and dividing the result by the trial duration.

For the simulations conducted and presented in Fig 6, the degree of oscillatory behavior exhibited by artificial HAs was also explored. The *mean peak frequency* (Hz) was computed by constructing a frequency power spectrum of each HA's angular position time series using MATLAB R2020's *pwelch* function. The mean peak frequency was defined by the frequency with the most power between 0.2 to 2 Hz and averaged across the artificial HAs. A Hamming window of 1024 samples with a 50% window overlap was used in constructing the frequency power spectrum.

## Procedure

Following informed consent, participants were taken to the testing room and received task instructions. Dyads were told that they had 45 minutes to complete eight successful trials of the corralling task. The experimented ended either when 45 minutes elapsed, or when eight successful trials were completed (whichever came first). Although trial success was determined during the last 90 s, participants were told to keep the TAs (referred to as "sheep" to participants) sufficiently contained for as long as possible. The experimenter clarified to participants that there was no pre-defined location for where the TAs should be contained–only that the TAs should be kept together somewhere on the grass field. Additionally, participants were informed that they could not communicate during the task, or during any breaks. The experimenter in the room enforced this no-talking policy.

## Model simulations

Simulations using the same task environment as presented to human participants (excluding the avatars) were conducted with two artificially controlled HAs implementing the task-dynamic model presented in Eqs 1 and 2 using the TA selection rule defined by Eq 3 in **Target agent selection** below. The simulations were conducted using Unity (ver. 2017.4.40f, Unity Technologies, San Francisco, USA). The behaviors of the artificial HAs were manipulated by setting the stiffness, $\varepsilon$, and damping, $b$ (via the damping ratio, $\zeta$), parameters for both Eqs 1 and 2. Relative to a given $\varepsilon$, a system can be described as under-/over-damped via the damping ratio, $\zeta$, which is the ratio between the value of $b$ and its critical value, which is equal to $2\sqrt{m\varepsilon}$ (where $m = 1$ kg). When $\zeta < 1$, the system will intersect the attractor in less time but will overshoot. When $\zeta > 1$, the system will approach the attractor slowly. When $\zeta = 1$, the system will approach the attractor with the least amount of time that results in at most one overshoot [68]. By varying $\zeta$ to define $b$, it is possible to assess the effect of underdamped or overdamped dynamics on the resultant patterns of behavior during the corralling task as a function of $\varepsilon$. In addition to setting stiffness and damping, the minimal radial distance to a selected TA, $r_{min}$, in Eq 1 was also varied.

A summary of the parameters that were considered is presented in **Table 1**. At each time-step, each artificial HA selected a TA to pursue using the selection rule defined in Eq 3 (see **Target agent selection** below, $\Delta t = 1$). The artificial HAs completed five trials with each parameter combination, and trials were 120 s in duration. The dynamics governing the HA and TAs behaviors, including the TA selection rule, were updated at 50 Hz. The simulation speed was set to 100 times faster than real time.

**Target agent selection.** In a corralling task with multiple TAs ($TA_k$; where $k = 1, 2, \ldots, M$), $HA_i$ is hypothesized to implement the intuitive rule of selecting and moving towards the TA that is (i) farthest from the task goal [11, 12, 17] and is (ii) moving away from the goal. These rules can be expressed as follows:

$$\{TA_{(t),i} \in H | \max(\|\boldsymbol{x}^{TA_{(t)i}} + \dot{\boldsymbol{x}}^{TA_{(t)i}}\Delta t\|)\}, \tag{8}$$

where,

$$H = \{TA_1, TA_2 \ldots TA_M\}, \tag{9}$$

and $\boldsymbol{x}^{TA_{(t)i}}$ and $\dot{\boldsymbol{x}}^{TA_{(t)i}}$ represents the position and velocity vectors, respectively, of $TA_{(t),i}$ in relationship to $\mathbb{C}$ in Cartesian space. The value $\dot{\boldsymbol{x}}^{TA_{(t)i}}\Delta t$ represents the positional increment added to $\boldsymbol{x}^{TA_{(t)i}}$ at time $t$ to predict $TA_{(t),i}$'s position at $t+\Delta t$. To summarize, an HA is hypothesized to select the TA at time $t$ that will be farthest from $\mathbb{C}$ at time $t+\Delta t$.

In the presence of multiple HAs, $HA_i$ only considers TAs that are closer to themselves than to their co-actors. For two HAs, this can be understood as both HAs ($i$ and co-actor $j$) creating a boundary, $\bar{\mathbb{B}}$, that is the perpendicular bisector of the line between them, $\boldsymbol{x}^{HA_i}\overline{\boldsymbol{x}}^{HA_j}$, at their mean position $\bar{\boldsymbol{x}}^{HA}$ (see Fig 2). Using point-slope form, where the slope, $m$, of $\bar{\mathbb{B}}$ is $-\frac{(x_1^{HA_i} - x_1^{HA_j})}{(x_2^{HA_i} - x_2^{HA_j})}$,

**Table 1. Model parameters adjusted for simulations.**

| Parameter | Start Step | End Step | Step Size |
|---|---:|---:|---:|
| $\sqrt{\varepsilon} = \sqrt{\varepsilon_r} = \sqrt{\varepsilon_\theta}$ | 2.5 | 10 | 0.25 |
| $\zeta = \frac{b = b_r = b_\theta}{2\sqrt{\varepsilon}}$ | 0.5 | 2 | 0.05 |
| $r_{min}$ | 0.3 | 0.4 | 0.05 |

the subset of TAs assigned to a given HA can be expressed as the following,

$$H_i = \{TA_{(t),i} \in H|f\}, \tag{10}$$

where $H_i$ is the subset of TAs assigned to $HA_i$ and $f$ is the following step function,

$$f = \begin{cases} x_2^{TA_{(t)}} - \bar{x}_2^{HA} > m(x_1^{TA_{(t)}} - \bar{x}_1^{HA}), & if \ \boldsymbol{x}^{HA_i} > \bar{\bar{\mathbb{B}}} \\ x_2^{TA_{(t)}} - \bar{x}_2^{HA} < m(x_1^{TA_{(t)}} - \bar{x}_1^{HA}), & if \ \boldsymbol{x}^{HA_i} < \bar{\bar{\mathbb{B}}} \end{cases}. \tag{11}$$

To summarize, Eq 11 assigns TAs to $H_i$ that fall on the same side of the boundary $\bar{\bar{\mathbb{B}}}$ as $HA_i$. Finally, $H_i$ replaces $H$ in Eq 9 to form the resultant TA selection rule for $HA_i$,

$$\{TA_{(t),i} \in H_i | \max(\|\boldsymbol{x}^{TA_{(t)i}} + \dot{\boldsymbol{x}}^{TA_{(t)i}} \Delta t\|)\}, \tag{12}$$

where Eq 12 is the same as Eq 3. To summarize, in task contexts with multiple HAs and TAs, the TA selection rules described above ensure that each $HA_i$ will pursue the TA in its assigned subset that is both nearest to $HA_i$ at time $t$ and predicted to be the farthest from $\mathbb{C}$ at time $t$ $+\Delta t$. Although only two HAs were considered in the simulations presented here, these selection rules can be extended to larger HA groups.

## Supporting information

**S1 Fig. A modified version of Fig 6.** The gray areas in the modified figure represent the 95% CI of human mean locomotion speed (HA Mean Speed, top row), human Cumulative $2\pi$ Rotations (middle row) and TA Mean Speed during the human experiment (bottom row). The gray areas are displayed separately for the low- (left column), medium- (middle column) and high-speed conditions (right column).
(TIF)

**S1 Video. Representative example of human participant behavior in the low-speed condition.** The video shows the participants' movements in the real environment, and how those movements translated in the virtual task environment. The video also includes each participant's view from their perspective when completing the task.
(MP4)

**S2 Video. Representative example of human participant behavior in the medium-speed condition.** The video shows the participants' movements in the real environment, and how those movements translated in the virtual task environment. The video also includes each participant's view from their perspective when completing the task.
(MP4)

**S3 Video. Representative example of human participant behavior in the high-speed condition.** The video shows the participants' movements in the real environment, and how those movements translated in the virtual task environment. The video also includes each participant's view from their perspective when completing the task.
(MP4)

**S4 Video. Example of emergent, oscillatory containment behaviors by artificial herding agent (HAs) ($\varepsilon = 64$, $b = 9.6$, $r_{min} = 0.35$, TA maximum speed $\leq 0.28$ m·s⁻¹).** For each HA, the pursued TA at time $t$ is color-coded to match the pursuing HA (i.e., blue or orange).
(MP4)

**S5 Video. Example of emergent, circling containment behaviors by artificial herding agent (HAs) ($\varepsilon = 49$, $b = 21$, $r_{min} = 0.35$, TA maximum speed $\leq 0.28$ m·s⁻¹).** For each HA, the

pursued TA at time $t$ is color-coded to match the pursuing HA (i.e., blue or orange).
(MP4)

**S6 Video. Example of emergent, solitary circling behavior by an artificial herding agent (HA) ($\varepsilon = 64$, $b = 9.6$, $r_{min} = 0.35$, TA maximum speed $\leq 0.28$ m·s$^{-1}$).** The pursued TA at time $t$ is color-coded (in orange) to match the pursuing HA.
(MP4)

## Acknowledgments

The authors thank Dr. Maurice Lamb for assisting in equipment management and setup.

## Author Contributions

**Conceptualization:** Patrick Nalepka, Paula L. Silva, Rachel W. Kallen, Kevin Shockley, Anthony Chemero, Michael J. Richardson.

**Data curation:** Patrick Nalepka.

**Formal analysis:** Patrick Nalepka.

**Investigation:** Patrick Nalepka.

**Methodology:** Patrick Nalepka, Paula L. Silva, Rachel W. Kallen, Kevin Shockley, Anthony Chemero, Elliot Saltzman, Michael J. Richardson.

**Resources:** Patrick Nalepka.

**Software:** Patrick Nalepka.

**Supervision:** Paula L. Silva, Rachel W. Kallen, Kevin Shockley, Anthony Chemero, Michael J. Richardson.

**Visualization:** Patrick Nalepka.

**Writing – original draft:** Patrick Nalepka.

**Writing – review & editing:** Patrick Nalepka, Paula L. Silva, Rachel W. Kallen, Kevin Shockley, Anthony Chemero, Elliot Saltzman, Michael J. Richardson.

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
