## [Decision Letter · Decision Letter 0]

14 Oct 2021

PONE-D-21-23404Task Dynamics Define the Contextual Emergence of Human Corralling BehaviorsPLOS ONE

Dear Dr. Nalepka,

Thank you for submitting your manuscript to PLOS ONE. After careful consideration, we feel that it has merit but does not fully meet PLOS ONE’s publication criteria as it currently stands. Therefore, we invite you to submit a revised version of the manuscript that addresses the points raised during the review process.

Specifically, the reviewers find value in the work, but one reviewer questions the connection between the experiment and simulations.  A revision of the manuscript may be able to address this issue.

We look forward to receiving your revised manuscript.

Kind regards,

Bryan C Daniels

Academic Editor

PLOS ONE

Journal Requirements:

"Dr. Nalepka was supported by the Macquarie University Research Fellowship. Prof. Chemero was supported by the Charles Phelps Taft Research Center at the University of Cincinnati. Prof. Richardson was supported by the Australian Research Council Future Fellowship (FT180100447). A/Prof. Kallen, Prof. Saltzman and Prof. Richardson were supported by the National Institutes of Health (R01GM105045). We thank Dr. Maurice Lamb for assisting in equipment management and setup"

"PN was supported by the Macquarie University Research Fellowship. AC was supported by the Charles Phelps Taft Research Center at the University of Cincinnati. MJR was supported by the Australian Research Council Future Fellowship (https://www.arc.gov.au/) (FT180100447). RWK, ES and MJR were supported by the National Institutes of Health (https://www.nih.gov/) (R01GM105045). The funders had no role in study design, data collection and analysis, decision to publish, or preparation of the manuscript"

Reviewers' comments:

Reviewer's Responses to Questions

**Comments to the Author**

1. Is the manuscript technically sound, and do the data support the conclusions?

Reviewer #1: Partly

Reviewer #2: Yes

2. Has the statistical analysis been performed appropriately and rigorously? 

Reviewer #1: Yes

Reviewer #2: Yes

3. Have the authors made all data underlying the findings in their manuscript fully available?

Reviewer #1: Yes

Reviewer #2: Yes

4. Is the manuscript presented in an intelligible fashion and written in standard English?

Reviewer #1: Yes

Reviewer #2: Yes

5. Review Comments to the Author

Reviewer #1: 1. The paper is well written and attempts to make a connection between human behavior in encirclement experiments in humans and a mathematical model that is supposed to explain this behavior. However, this connection is presented in Fig.6 and it is unclear:

- As explained in the text and Fig. 5 there are two qualitatively different behaviors that appear in the simulations: circling movements (overdamped) and oscillatory behavior (underdamped). If the gray areas in Fig. 6 represent the experiments, it seems that most of the cases are in the overdamped region, so where is the experimental justification for the existence of both behaviors?

- The results of the simulations and experiments are mapped using HA mean speed, HA cumulative 2Pi roations or Mean Travel? According to the caption of the figure only HA mean speed is considered (?). All the three could be used and it will be important to check if the experimental data fall in the same region of the Damping-Stiffness coordinates. Otherwise the connection between this model and the experimental data is not convincing enough and it is supposed to be one of the main pillars of this study.

2. If only Eqs. 1,2 and 11 are relevant to the model, they should appear together. Eqs [Disp-formula pone.0260046.e019]–[Disp-formula pone.0260046.e024] which are not part of the model should appear only in the discussion (where they are relevant).

3. Is there a dependence of the results (experiments and simulations) on the number of TAs? As I understand that only 7 TAs were considered.

Reviewer #2: Although I find myself outside of the particular scientific discipline explored in this paper, with my insufficient grasp of long equations, I believe I understood the major conceptual advance and still hope the editor finds my review useful.

Altogether, the paper reads as an excellent contribution to the literature. With highly impressive methods, and important conclusions. Specifically, the authors placed naive human dyads in a three dimensional virtual world using VR headsets, and asked them to corral a group of seven agents (I like to think of them as "prey"). They found that humans performed circling behaviours around the prey while keeping their partner as close to the opposite side of the circle as possible. The really critical component is that the authors were able to replicate some (although not all) aspects of this behaviour using similar task dynamic equations to their original set of papers. This suggests human minds may be running calculations similar to those suggested here. Does this suggest that early hunter gatherers were circling their prey? Or do we just have enough flexibility in our cognition to solve this task cognitively? The authors are, in my opinion, welcome to explore these questions or just ignore them.

I think a summary paragraph would benefit this paper (again at authors discretion). Further than that, I have no major or minor comments and believe the paper is ready for publication in its present form. It read like a published work, and I must commend the authors on their clarity.

Daniel W. E. Sankey

(I sign all my reviews)

6. PLOS authors have the option to publish the peer review history of their article (what does this mean?). If published, this will include your full peer review and any attached files.

Reviewer #1: No

Reviewer #2: **Yes: **Daniel W. E. Sankey

---

## [Editor Report · Decision Letter 1]

2 Nov 2021

Task dynamics define the contextual emergence of human corralling behaviors

PONE-D-21-23404R1

Dear Dr. Nalepka,

We’re pleased to inform you that your manuscript has been judged scientifically suitable for publication and will be formally accepted for publication once it meets all outstanding technical requirements.

Kind regards,

Bryan C Daniels

Academic Editor

PLOS ONE
---

## [Editor Report · Acceptance letter]

5 Nov 2021

PONE-D-21-23404R1 

Task dynamics define the contextual emergence of human corralling behaviors 

Dear Dr. Nalepka:

I'm pleased to inform you that your manuscript has been deemed suitable for publication in PLOS ONE. Congratulations! Your manuscript is now with our production department. 

Kind regards, 

on behalf of

Dr. Bryan C Daniels 

Academic Editor

PLOS ONE